# Mapping potential connections between Southern Africa's elephant populations

**Ryan M. Huang**[1‡]*, **Rudi J. van Aarde**[2‡]*, **Stuart L. Pimm**[1,2], **Michael J. Chase**[3], **Keith Leggett**[4]

**1** Nicholas School of the Environment, Duke University, Durham, North Carolina, United States of America, **2** Conservation Ecology Research Unit, Department of Zoology and Entomology, University of Pretoria, Hatfield, South Africa, **3** Elephants Without Borders, Kasane, Botswana, **4** Fowlers Gap Arid Zone Research Station, UNSW Sydney, Sydney, Fowlers Gap, Australia

‡ RMH and RJA are joint senior authors on this work.
* ryan.huang@duke.edu (RMH); rjvaarde@zoology.up.ac.za (RJA)

## Abstract

Southern Africa spans nearly 7 million km$^2$ and contains approximately 80% of the world's savannah elephants (*Loxodonta africana*) mostly living in isolated protected areas. Here we ask what are the prospects for improving the connections between these populations? We combine 1.2 million telemetry observations from 254 elephants with spatial data on environmental factors and human land use across eight southern African countries. Telemetry data show what natural features limit elephant movement and what human factors, including fencing, further prevent or restrict dispersal. The resulting intersection of geospatial data and elephant presences provides a map of suitable landscapes that are environmentally appropriate for elephants and where humans allow elephants to occupy. We explore the environmental and anthropogenic constraints in detail using five case studies. Lastly, we review all the major potential connections that may remain to connect a fragmented elephant metapopulation and document connections that are no longer feasible.

## Introduction

The UN has declared 2021–2030 the 'decade of restoration' (https://www.decadeonrestorati on.org), an initiative that aspires to many actions. They must include reconnecting nature [1, 2]. Even when natural habitats remain and are protected, they are often small, isolated, and unable to sustain viable populations [3]. Human activities surround habitats with unsuitable areas or constrict animals' movements with artificial barriers, such as roads or fences [4].

Reconnecting such fragments is an obvious solution for the conservation benefits of connecting protected areas are numerous [5]. In Africa, connections follow decades of effort to reduce illegal harvesting and habitat loss, partially by buffering protected areas with conservation-sensitive activities on lands that surround them [6, 7]. The resulting connectedness and enlargement of protected estates agree with the guiding principles of the Global Deal for Nature (GDN) to enhance the connectivity of protected areas to improve population viability and persistence, especially of large-bodied herbivores that range over wide areas [1]. The

**Data Availability Statement:** Telemetry data cannot be shared publicly because they contain the locations of an endangered and highly poached species. The environmental and human data necessary for reproducing our results are available

at: https://osf.io/eyanr/. Much of these data have also been published under Loarie et al (2009).

**Funding:** Support for this study was provided by Billiton, Conservation Foundation Zambia, Conservation International's southern Africa's Wildlife Programme, the Conservation Lower Zambezi, the International Fund for Animal Welfare, the Mozal Community Development Trust, the National Research Foundation, the National Postcode Lottery of the Netherlands, Peace Parks Foundation, the US Fish and Wildlife Services, the University of Pretoria, the World Wildlife Fund (SARPO; Mozambique; SA), the Walt Disney Grant Foundation, and the Wildlifewins Lottery. Elephants Without Borders was funded by the Paul G. Allen Family Foundation, Jody Allen, the Woodtiger Fund, the Thomas C Bishop Charitable Foundation, the James and Deborah Burrows Foundation, and the Zoological Society of San Diego. We acknowledge the in kind logistical support of South African National Parks. This research was sanctioned and supported by the Botswana Dept. of Wildlife & National Parks, Direcção Nacional de Areas de Conservação, the Namibian Ministry of Tourism & Environment, the Malawian Wildlife Dept., the South African National Parks, Ezemvelo KZN Wildlife of South Africa, and the Zambian Wildlife Authority. We also thank the Gonarezhou Conservation Trust for telemetry data from Gonarezhou National Park. Ifaw continues to support our ongoing research initiatives. The funders had no role in study design, data collection and analysis, decision to publish, or preparation of the manuscript.

**Competing interests:** The authors have declared that no competing interests exist.

practicality of connections surely depends on their extent. Small, local corridors show success [2, 8]. The promotion of continent-wide initiatives [9] begs to resolve the many details needed to implement them. It is such details we consider here.

Southern Africa (Angola, Botswana, Malawi, Mozambique, Namibia, South Africa, Eswatini, Zambia, and Zimbabwe) spans nearly 7 million $km^2$ and contains approximately 80% of the world's savannah elephants (*Loxodonta africana*) [10] mostly living in protected areas. A generous portion (~900,000 $km^2$) of land has been set aside for conservation. While it includes some of the world's largest protected areas, their isolation nonetheless creates problems. Here, using elephants as the focal species, we consider the prospects for improving the connections between the region's remaining populations [11].

Our overarching goal is to map where elephants might be able to move between their current populations and, conversely, where they cannot. To achieve this, we use a long-term telemetry dataset of elephant movements throughout southern Africa to identify how elephants utilize a landscape. We have four major sections, with accompanying objectives.

Our first specific objective considers "where do elephants want to go." We combine the telemetry data with the substantial literature on the environmental constraints on elephant movement. For instance, despite the wide range of elephants, their movement depends on the availability of water sources [12, 13]. Additionally, though catholic in their feeding habitats, elephants prefer some habitats over others [14–16]. Lastly, there are natural barriers to movement, such as deep or fast-flowing rivers or steep mountainsides.

Second, in the section "How human actions restrict elephant movements," we consider barriers to movements. Again, using telemetry data as a guide, we identify human factors, including population, agriculture, and fencing, that prevent or restrict elephant dispersal across the otherwise suitable habitat. The presence of people (and their crops and livestock) at high densities is an obvious example of such limits [17]. Even in areas of low human density, elephant movement through cropland is undesirable as it leads to human-elephant conflict [18]. Fencing protected areas is a tempting solution to conflicts but problematic [19] and hotly debated [20–22]. We explore these issues further in the Discussion.

Our third objective recognises that while these natural and human-imposed factors provide overarching constraints, they vary in their significance across landscapes. Through five regional case studies, we demonstrate how different local environmental and human factors work in combination to direct elephant movement.

Our fourth objective is to describe specific routes of connectivity that remain between elephant populations and their suitability given the constraints we have described. We summarise knowledge to see where reconnecting protected landscapes may afford elephants with dispersal opportunities and ultimately restore the metapopulation [11]. Where are the existing fences? Are conservation actions possible, or must managers prepare for the problems that isolated populations with restricted movement may cause? Ideally, connectivity should safeguard populations in protected areas from the consequences of isolation [17, 23, 24] by allowing migration [25], and mitigating the negative impacts that elephants may have on other species [26].

## Methods

### Telemetry data

We assembled location data on 261 elephants from 1993 to 2018 (S1 Table), including 1.2 million observations from eight southern African countries (Angola, Botswana, Malawi, Mozambique, Namibia, South Africa, Zambia, and Zimbabwe). These data demonstrate how elephants respond to environmental and human factors across a landscape. Although elephant movements vary seasonally [25], we wanted to map all areas that may allow for connectivity,

even if only for a portion of the year. The telemetry collaring was subject to ethical review and was approved by the Animal Ethics Committee of the University of Pretoria (AUCC-040611-013) and also approved by the Botswana Ministry of Environment, Wildlife, & Tourism (OP 46/1 LXXXV 89).

We assigned elephant data to eight clusters of protected areas: Etosha (Namibia), Chobe (northern Botswana, Zambezi region of Namibia (formerly called the Caprivi Strip), and parts of north-west Zimbabwe), Kafue (Zambia), Limpopo, which includes Kruger National Park (South Africa, Zimbabwe, and Mozambique), Luangwa (Zambia and Malawi), Maputo (southern Mozambique and South Africa), Niassa (Mozambique), and Zambezi (Zambia and Zimbabwe) [27]. These are areas where primary protected areas (IUCN categories I-IV) form core conservation areas surrounded by and connected to secondary protected areas (IUCN categories V-VI) that may act as conservation buffers. Most of southern Africa's elephants live in these primary protected areas, but what else explains where elephants prefer to live when given the opportunity?

## Environmental and anthropogenic data

To map environmental and anthropogenic constraints on elephant distributions, we combined GIS layers from several sources. For each data layer, we calculate the distribution for all telemetry points and identify a threshold value of elephant preference. We used a 90m digital elevation map of slope [28], the Southern Africa Regional Science Initiative Project map of vegetation [29], and the HydroRIVERS dataset to map river locations [30]. HydroRIVERS classify rivers by flow order, based on a log-10 scale of long-term average discharge. Rivers in our study extent range from 3$^{rd}$ order, with an annual flow discharge of $>1,000$ m$^3$ s$^{-1}$, down to 7$^{th}$ order, with $>0.1$ m$^3$ s$^{-1}$.

When modelling the influence of humans on the landscape, we used the LandScan 2019 1 km human population density data [31] and the Gridded Livestock of the World 5 minute map of cattle distribution from the Food And Agriculture Organization (FAO) [32]. As a proxy for agriculture, we used a dataset developed using a neural network and one million data points to predict cropland locations in sub-Saharan Africa at a 1 km resolution [33]. We derive our map of fence locations from a combination of existing GIS data from the FAO [34] and hand digitization using satellite imagery and published literature [4].

## Suitable landscapes

Once we understand the individual effects of the environment and human activity on perceived elephant preferences, we may synthesise these insights to map what areas are left to elephants. Our analysis covers all eight countries for which we had telemetry data for, and for Eswatini, which has a small elephant population. We exclude Lesotho given the absence of elephants. Given the preference thresholds we identified using the telemetry data, we start with binary layers of preferred or unpreferred land (Fig 1). We then combine the environmental spatial layers (vegetation, slope, and distance to rivers) in a subtractive manner, leaving only the common areas across the layers. We repeat this process for the anthropogenic layers (human density, cattle density, and crop probability) but here, we added protected areas with an IUCN classification or equivalent of I-VI (Fig 1F) back in [35]. The was because the data on cattle presence were aggregated at the county level and sometimes predicted elevated cattle densities within protected area boundaries. Adding protected areas back represents our capacity to manage and mitigate detrimental effects of human activity within protected areas. Lastly, we identify the overlap between areas that are environmentally suitable for elephants and areas that humans allow elephants to occupy (S2 Table, S1 Fig).

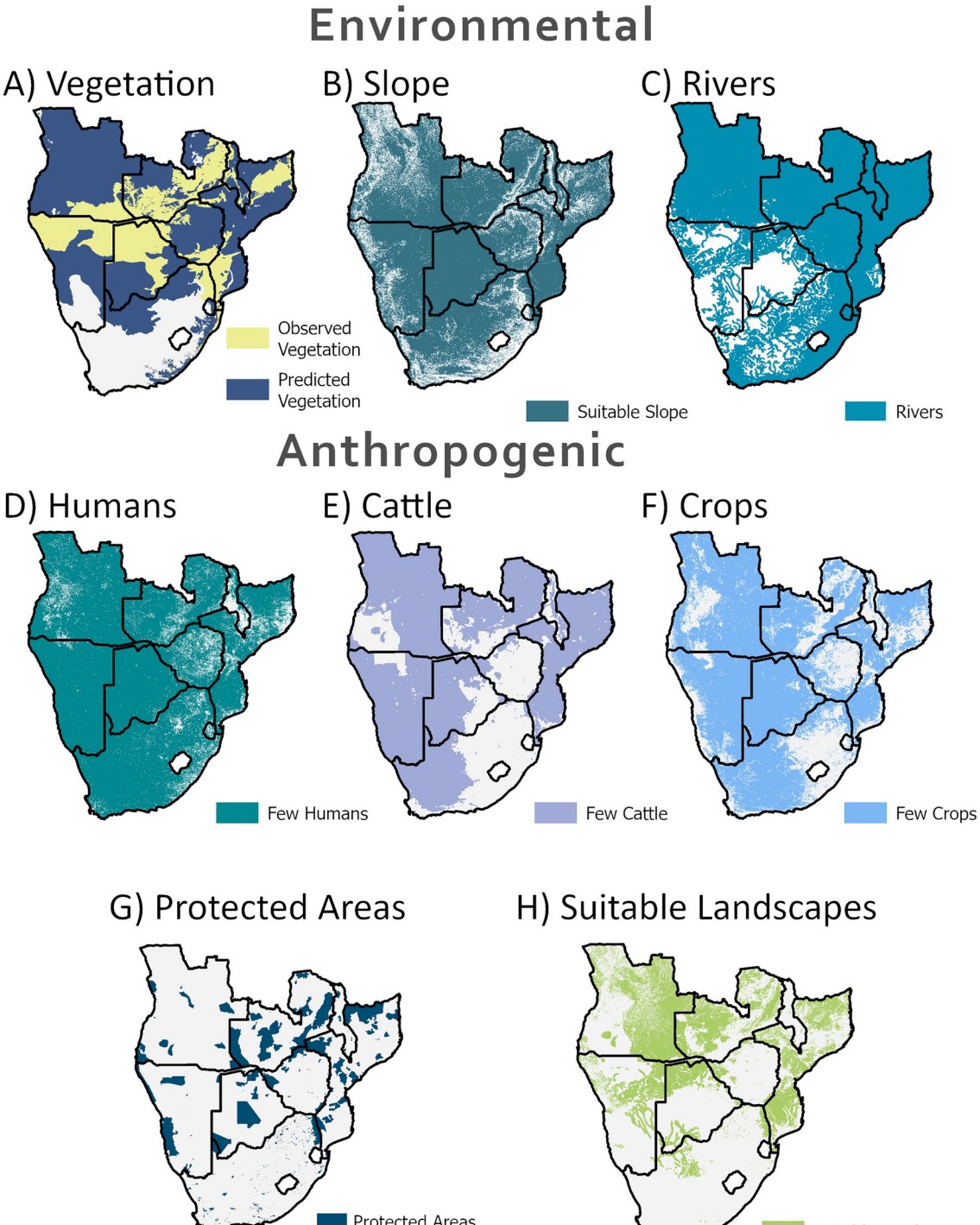

**Fig 1. Composite of suitable areas for elephants for individual spatial layers.** Areas suitable for elephants are calculated for each geospatial data separately. These include vegetation (A) [29], slope (B) [28], distance to rivers (C) [30], human population (D) [31], cattle density (E) [32], crop probability (F) [33], and protected areas (IUCN I-VI) (G) [35]. The intersection of these seven layers provides our projection of suitable landscapes for elephants (H).

## Results

### Where do elephants want to go?

**Vegetation.** The Southern Africa Regional Science Initiative Project [29] has mapped 252 different vegetation zones across this region. Of these, our telemetry data occur in 83 (shown in yellow, Fig 1A). After identifying the tree species in these 83 zones, we predict an additional 76 vegetation zones with the same species that might be suitable for elephants (shown in blue, Fig 1A). Strongly associated species include mopane (*Colophospermum mopane*), silver cluster leaf (*Terminalia sericeamopane*), and russet bushwillow (*Combretum hereroense*).

**Slope.** Elephants avoid steep slopes [36]. Such behaviour is unsurprising for animals as large as elephants. Traversing these areas is likely to be energetically costly [37]. Our data show that elephants across all clusters prefer flat ground, with 95.8% of recorded presences on terrain <3˚ (S2 Fig). Slope's influence on movement is evident when looking at hilly regions such as Nyika National Park of Malawi and the Lower Zambezi National Park in Zambia (see regional case studies below). Telemetry paths of tracked elephants in these areas illustrate how hills act as barriers and restrict movement to valleys and other low-lying plains.

**Rivers.** Across southern Africa, elephants occur from the deserts of Namibia in the west to the evergreen moist forests of Mozambique in the east. This region contains some of Africa's largest rivers: the Zambezi, Luangwa, Kafue, Okavango, Limpopo, and Orange, and their tributaries. These water sources vary in their availability both spatially and temporally, with 3rd order rivers (long term average discharge of >1,000 $m^3$ $s^{-1}$) such as the Chobe River flowing year-round, while 7th order rivers (average discharge of 0.1–1 $m^3$ $s^{-1}$) are more prevalent (S3 Fig) but run dry for more of the year. Perhaps unsurprisingly, we find elephants frequent rivers of all sizes (Fig 2). Across the entire study extent, 94.5% of elephant records lie within 10 km of a 7th order river or larger and thus is the threshold for which we model river access. Although elephant movement is more restricted during the dry season2, our goal was to identify areas that may be suitable, even if only for part of the year.

### How human actions restrict elephant movements

**Human population, croplands, and cattle grazing.** Elephants have likely always shared land with humans, but the human population growth and rapid land cover change combined with the recovery of elephant populations across the southern Africa over the last century has increased human-elephant conflict [38–40]. Using data on cropland probability [33], there is a strong relationship between areas with low agricultural potential (activity) and those where elephants travel. 96.3% of telemetry points are in locations where the probability of growing crops is <25%. This pattern remains if we only consider areas outside of IUCN category I-VI protected areas (94.2% of elephant presences beyond protected areas are in similarly human-free areas, S4 Fig). Elephants similarly prefer areas with low human populations (<25 people/$km^2$, S5 Fig) [31]. Although the pattern is less clear for cattle densities (S6A Fig) [32], we use a threshold of <5 cattle/$km^2$ given that the presence of elephants in higher densities of cattle only occur in areas that directly abut the protected areas (S6B Fig).

**Fences.** Throughout southern Africa, landowners and governments commonly employ fences to separate wildlife from livestock (Fig 3). Many of these fences either border or intersect protected areas with elephant populations. In nearly every case for which we have telemetry data, fences restrict elephant movement. There are only a few instances where the fences appear to have gaps that elephants move through. In areas without fences, such as the Luangwa valley in Zambia and Quirimbas National Park in Mozambique, elephants move more freely, even occasionally travelling beyond protected area boundaries (Fig 3).

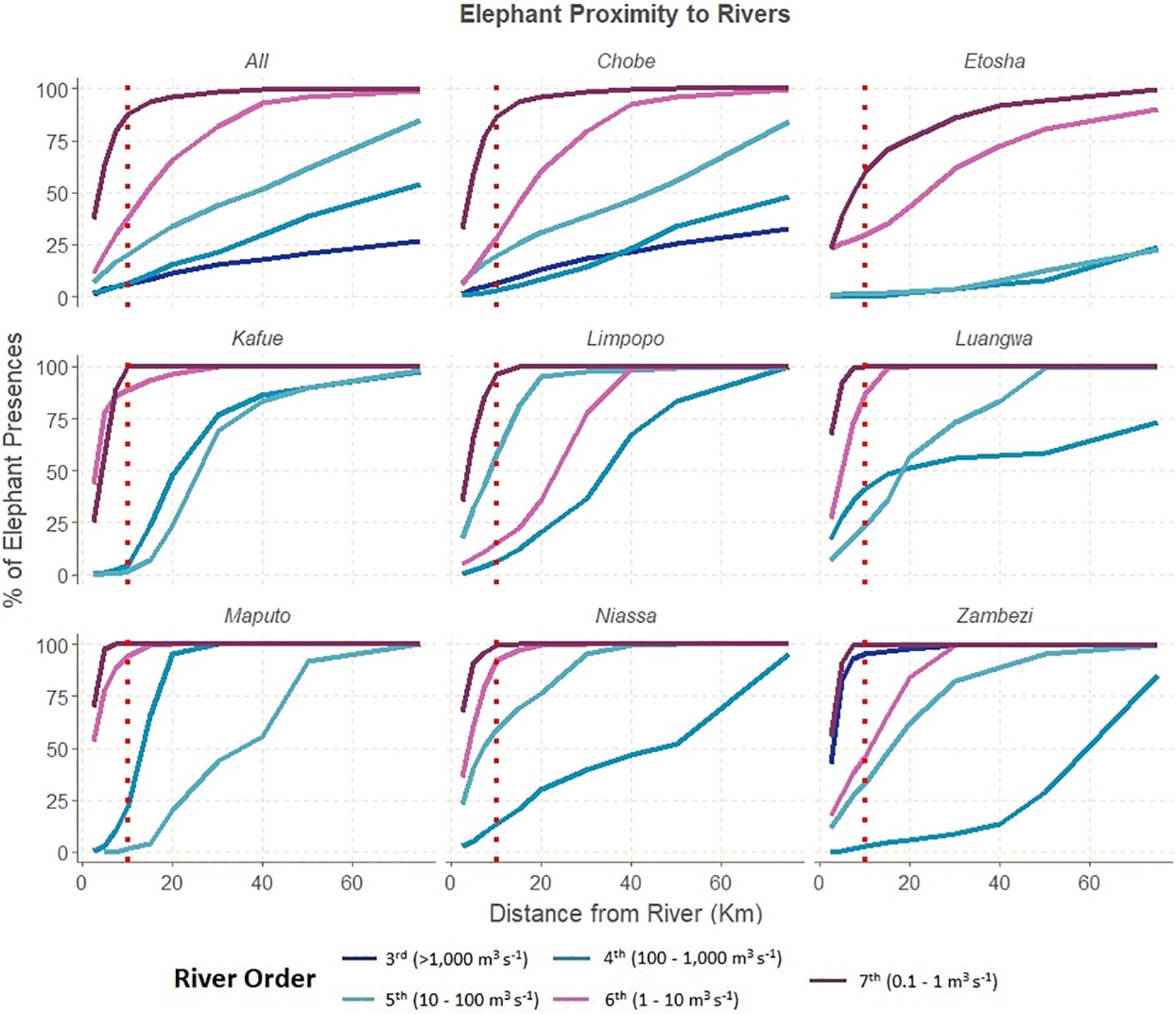

**Fig 2. Amount of land and elephant presences within distances of varying river sizes.** Accumulation curve of elephant telemetry points as the distance increases away from rivers of varying flow orders. The red dashed line indicates the 10 km preference threshold.

## Regional case studies

**Case 1: Etosha cluster.** The importance of water availability for elephants is evident in the most arid portion of their range (Fig 4). Etosha National Park and the Namibian interior receive approximately 350 mm of rain annually, whereas the coast only receives 50mm. Despite the harsh environment, elephants occur throughout, albeit in two distinct patterns. When inside the park, the availability of artificial watering holes permits relatively unrestricted movement, but paths outside the park shows that elephants have a strong affinity for riverbeds where they may dig for water. This pattern is clear the further west one travels towards the Skeleton Coast. Such fidelity in following riverbeds is unsurprising given the region's aridity and the dependence of elephants on water [41].

In addition to differences in water availability, fences affect movement here. The fences along the eastern half of Etosha are an effective barrier to dispersal, preventing individuals

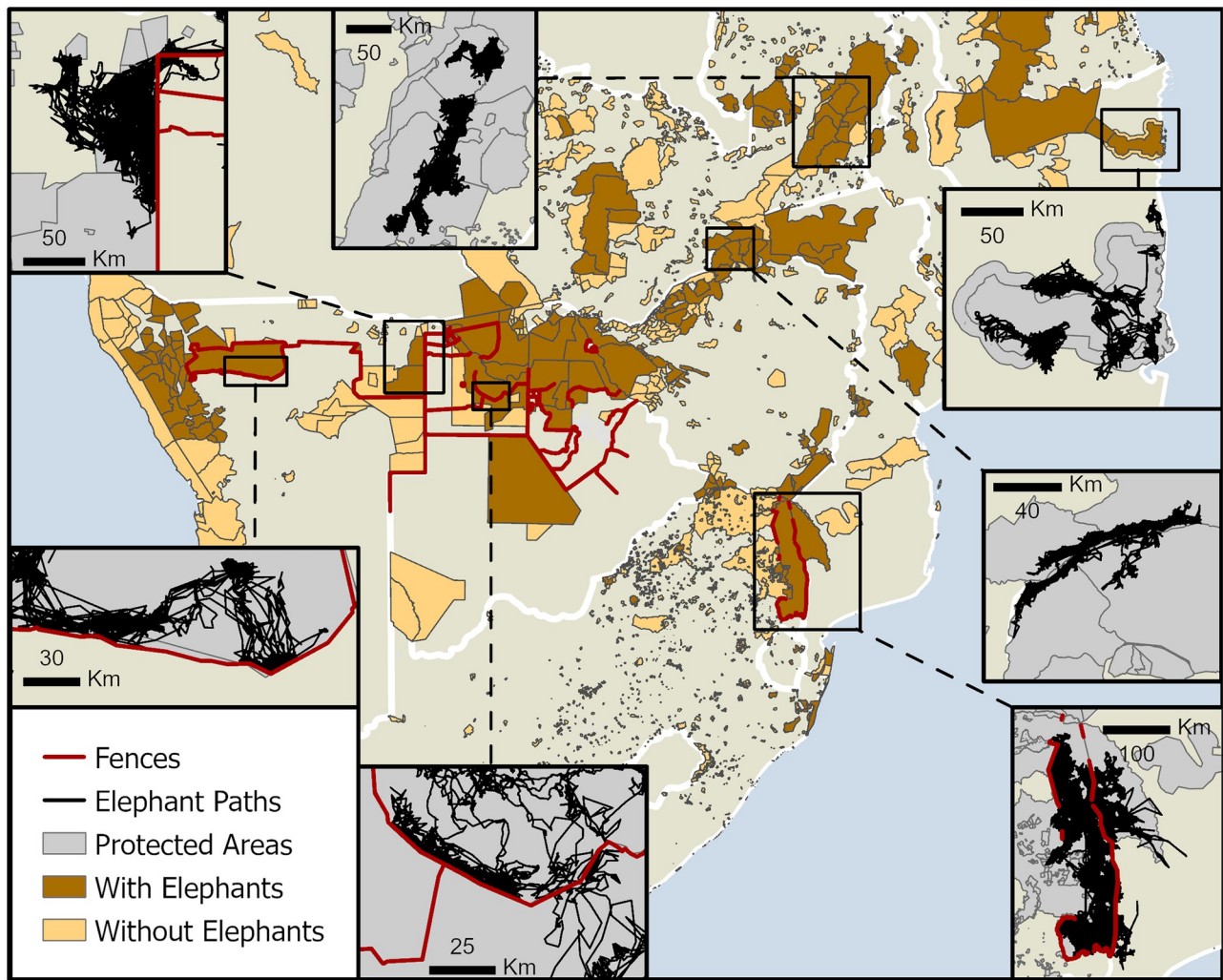

**Fig 3. Map of fences.** Fences (red) and elephant paths (lines between consecutive telemetry points) (black) across southern Africa. Protected areas (grey in inset maps) with elephants (dark brown) or without (light brown) appear on the large map.

from moving eastwards toward the elephant populations in Khaudum National Park and Botswana (Fig 4). Although managers have tried to keep fences intact, apparent gaps in the southwest corner of Etosha National Park allow elephants to move across the boundary.

These differences in movement patterns on either side of Etosha are practical cases illustrating where elephants would like to go and where humans allow them to. When provided the opportunity to move freely beyond protected areas, elephants will do so, even if it means dispersing into a drier and more resource-limited landscape.

**Case 2: Luangwa and Zambezi clusters.** Elephants along the Zambezi River and on the edge of the Luangwa valley in Zambia are less restricted by water availability than their conspecifics to the west, but instead are limited by the terrain (Fig 5). In the Lower Zambezi National Park, elephants move along the river floodplain and sometimes to the south, but rarely climb the hills to the north. In Vwaza Marsh Wildlife Reserve in Malawi, elephants are limited to the valley between the Nyika plateau and the foothills marking the Malawi-Zambia border. In both cases, the steep slopes apparently act as barriers that must be considered when mapping connectivity.

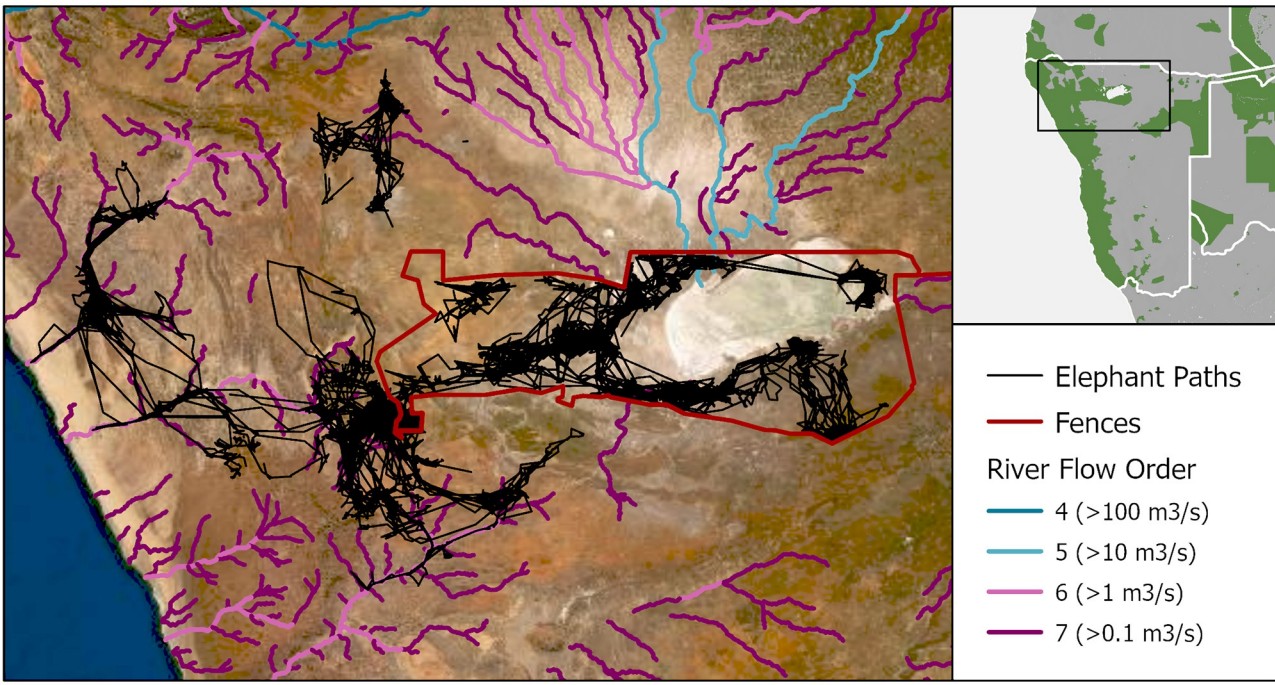

**Fig 4. Map of elephant paths in northern Namibia.** Fenced boundaries (red) [4] around Etosha National Park prevent elephant movement (black) towards the north, east, and south. The arid landscape restricts movement in the west to riverbeds (symbolised by average annual flow rates from blue to purple) [30]. Green areas in the subset map mark protected areas from the WDPA [35]. Basemap Source: ESRI, MAXAR.

**Case 3: Kasungu national park.** Telemetry data show elephants' strong preference for avoiding areas with high human density or agricultural probability, but there are exceptions. Such instances are usually cases of crop-raiding behaviour by elephants and illustrate the ongoing potential for human-elephant conflict. A clear example of this occurs just outside Kasungu National Park in Malawi (Fig 6). There is a very high concentration of human activity and settlements immediately outside the park's eastern boundary. However, ineffective fencing along the eastern boundary allows elephants to roam beyond the park limits and clash with local communities. Most of these excursions occur in the evenings (between 15:00–1:00 CAT), which is typical of elephants' crop-raiding behaviour [42].

**Case 4: Limpopo cluster.** Data from the Limpopo region provide a useful example of how we may start to connect isolated populations. Currently, elephant movements are confined to individual parks mainly due to a combination of anthropogenic activity and fences (Fig 7). This is most apparent to the west of Kruger National Park and north of Gonarezhou National Park. Conversely, we see more unrestricted movement to the east where there is less fencing and agricultural activity. Allowing such dispersal beyond country borders is a primary goal of transfrontier parks. Intended to connect Kruger in South Africa, Limpopo in Mozambique, and Gonarezhou in Zimbabwe, the Greater Limpopo Transfrontier Conservation Area would promote movement of individuals throughout the 80,000 km$^2$ region and connect these populations.

**Case 5: Zambezi region.** The eastern end of the Zambezi region of Namibia provides another example of the complex factors impeding elephant movements (Fig 8). In the south, fences separate elephants from areas of croplands. In the northeast, the main channel of the Zambezi River (coming in from the northwest) apparently prevents elephants from crossing [24]. For most of its length, the Chobe River is an effective barrier, but there are places where

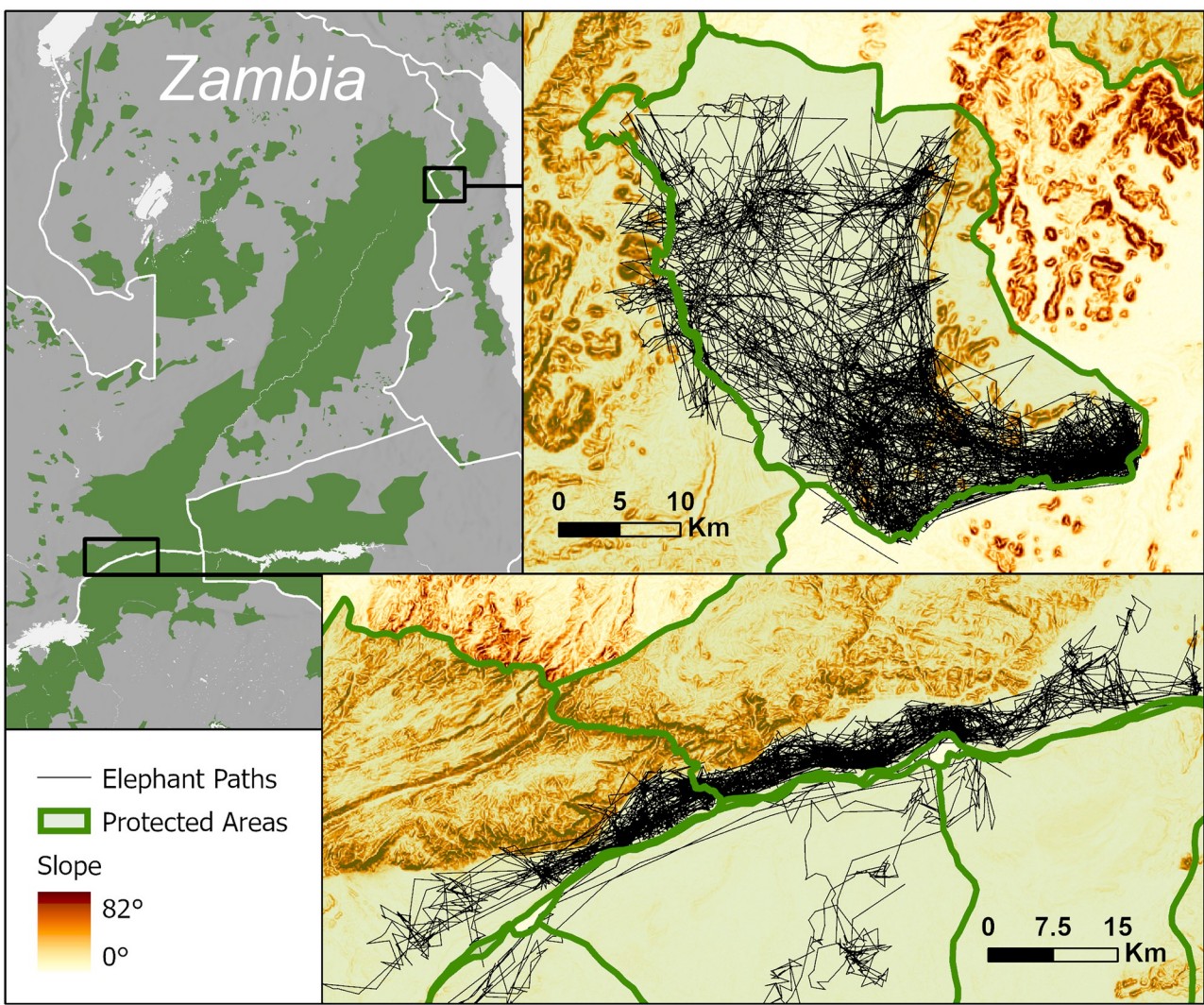

**Fig 5. Map of elephant paths in Luangwa and Zambezi.** Telemetry paths of elephants (black lines) in part of the Luangwa and Zambezi clusters overlaid on a map of slopes [28]. Green areas in the subset map mark protected areas from the WDPA [35].

elephants do cross in the west. Lastly, elephants navigate this human-dominated landscape and avoid areas with high human population density.

## What connections may still remain?

**Namibia, Botswana, and South Africa: The limitations of fences and water.** Fences are frequently employed throughout Namibia, Botswana, and South Africa to separate wildlife from domesticated animals (Fig 3). This extensive network crosses what would otherwise be vast uninterrupted swaths of suitable habitat (Fig 9), making fences the most significant barrier to connectivity in these countries. Furthermore, the aridity of the Namib desert and the Kalahari provide additional constraints on elephants' abilities to move through these regions. When we take a closer look at five potential connections in these countries, these limitations become apparent and how their removal may promote dispersal in some cases.

*Namibian Coast.* While Etosha National Park is intended to be fully fenced along its 824 km boundary, some of it is in a state of disrepair [43]. Gaps in the park's fences allow elephants

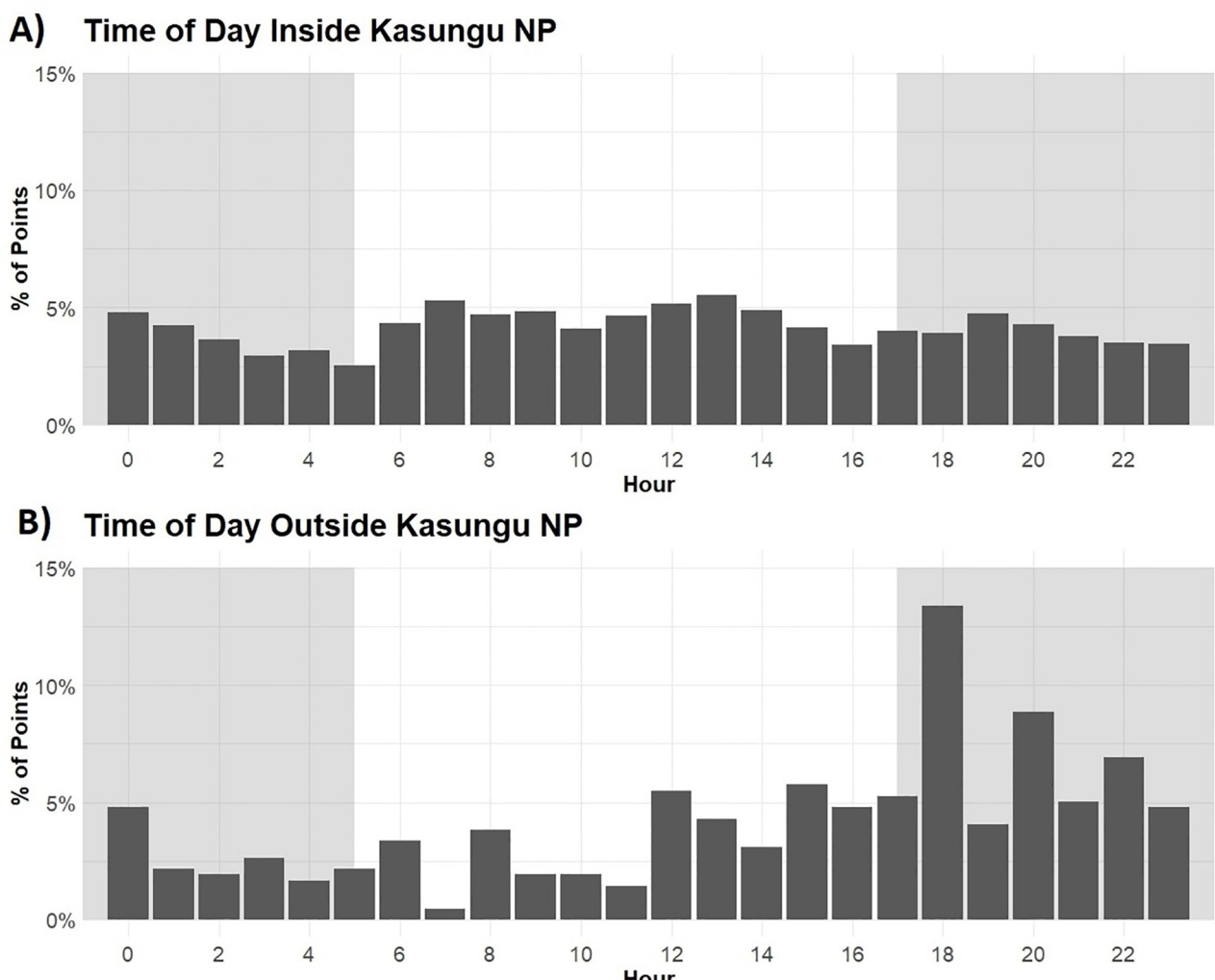

**Fig 6. Distribution of elephant presences by time inside and outside Kasungu National Park.** Histograms show elephant telemetry records' hourly distribution for elephants living in Kasungu National Park in Malawi. The data are split between those within the national park boundaries (A) and those outside (B). The shaded regions of the graphs approximately represent night (17:00–5:00). The difference in distributions suggests elephants more often appear outside the park at night, a behaviour typically associated with crop raiding.

to disperse to the west into areas nominally protected by communal conservancies (Fig 4). The establishment of these communal conservancies has generally been considered a success for wildlife conservation and tourism [44]. However, elephant activity has increased human-elephant conflict, particularly with pastoralists over water resources [45]. This is unsurprising given the intense aridity of the region. Because of the water scarcity, most movement is restricted to dry riverbeds where elephants must dig for water. Despite the limited suitable habitat, maintaining good relations with the communal conservancies may eventually link the elephant population within Etosha to the Namibe province of Angola.

*Northern Namibia.* When fences are upgraded and maintained, they are effective barriers against elephants, as is the case on Etosha's eastern boundary (Figs 3 and 4). Beyond the fences to the east and north of the park is commercial farmland, an area unsuitable for elephants. When elephants attempt to cross this region when moving westward from Khaudum National Park (Fig 3), they regularly conflict with farmers over water [46]. This land use interrupts any

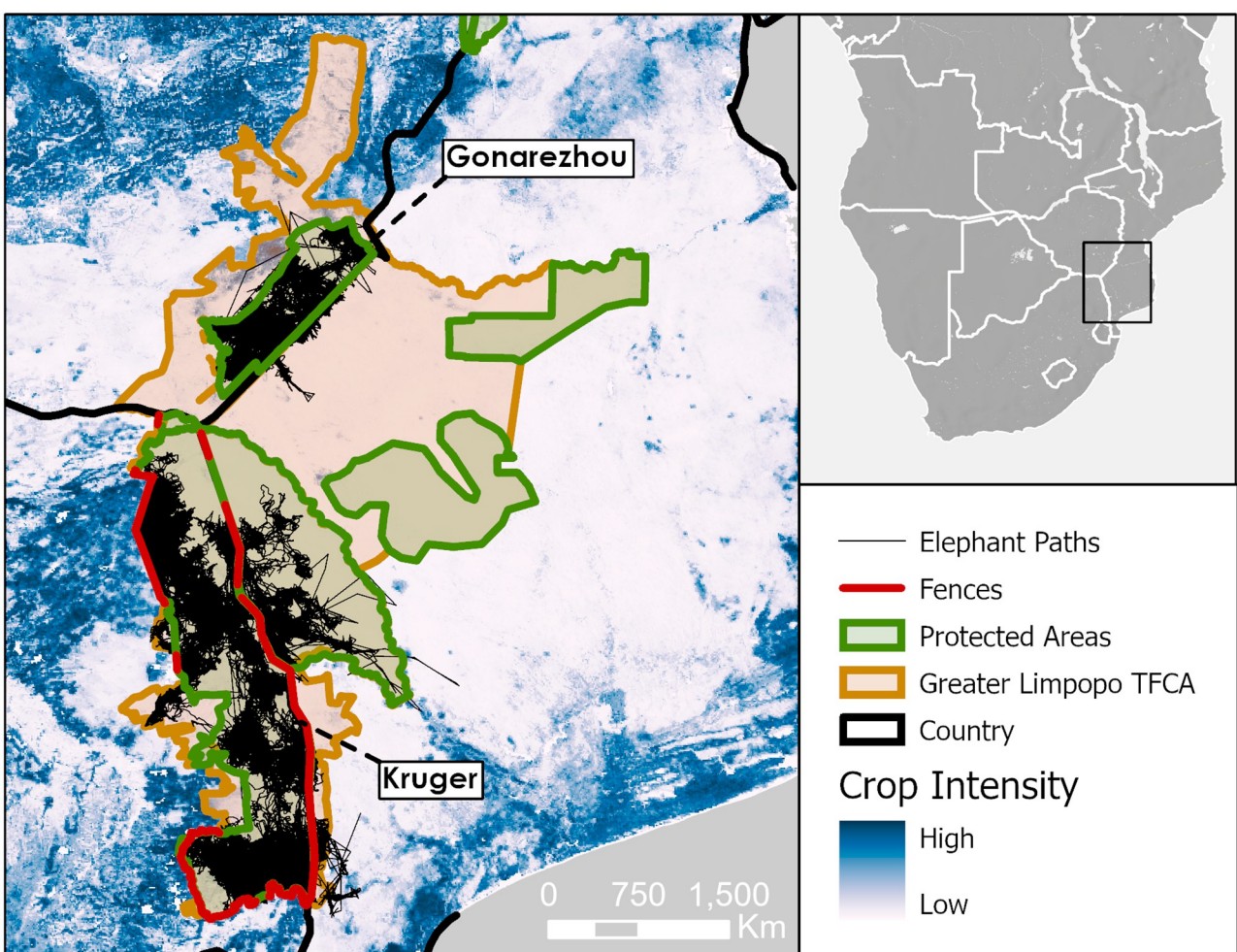

**Fig 7. Map of elephant paths and crop probability in the Limpopo cluster.** Telemetry paths of elephants in Kruger National Park (South Africa), the adjacent Limpopo National Park (Mozambique), and Gonarezhou National Park (Zimbabwe) overlaid on a map of crop probability (blue) [33]. Fences are marked in red [4], and the WDPA I-VI protected area boundaries are green [35]. The proposed Greater Limpopo Transfrontier Conservation Area is in orange.

potential connectivity between elephants in Etosha and the rest of the southern African sub-continent [17]. Without a feasible route to connect these populations, interventions to prevent or reduce human-elephant conflict, such as more effective fencing around Khaudum, may be a more reasonable management action for this area.

*Namibia-Botswana border*. Despite the Kavango-Zambezi Transfrontier Conservation Area's mission to facilitate transfrontier dispersal [47], the international border fence is a hard barrier to movement between Namibia and Botswana (Fig 3). Elephants in Khaudum cannot easily reach elephants on the western banks of the Okavango delta despite the proximity and suitable habitat. Instead, individuals from Khaudum may travel through the Zambezi region, but even this route faces difficulties crossing the Okavango River into Bwabwata National Park. While establishing the transfrontier conservation area is ongoing, prioritising the permeability of the international border fence might be considered to achieve connectivity in the region [47].

*Central Kalahari*. Although not part of the Kavango-Zambezi Transfrontier Conservation Area, ongoing tracking efforts indicate that elephants are actively moving between the

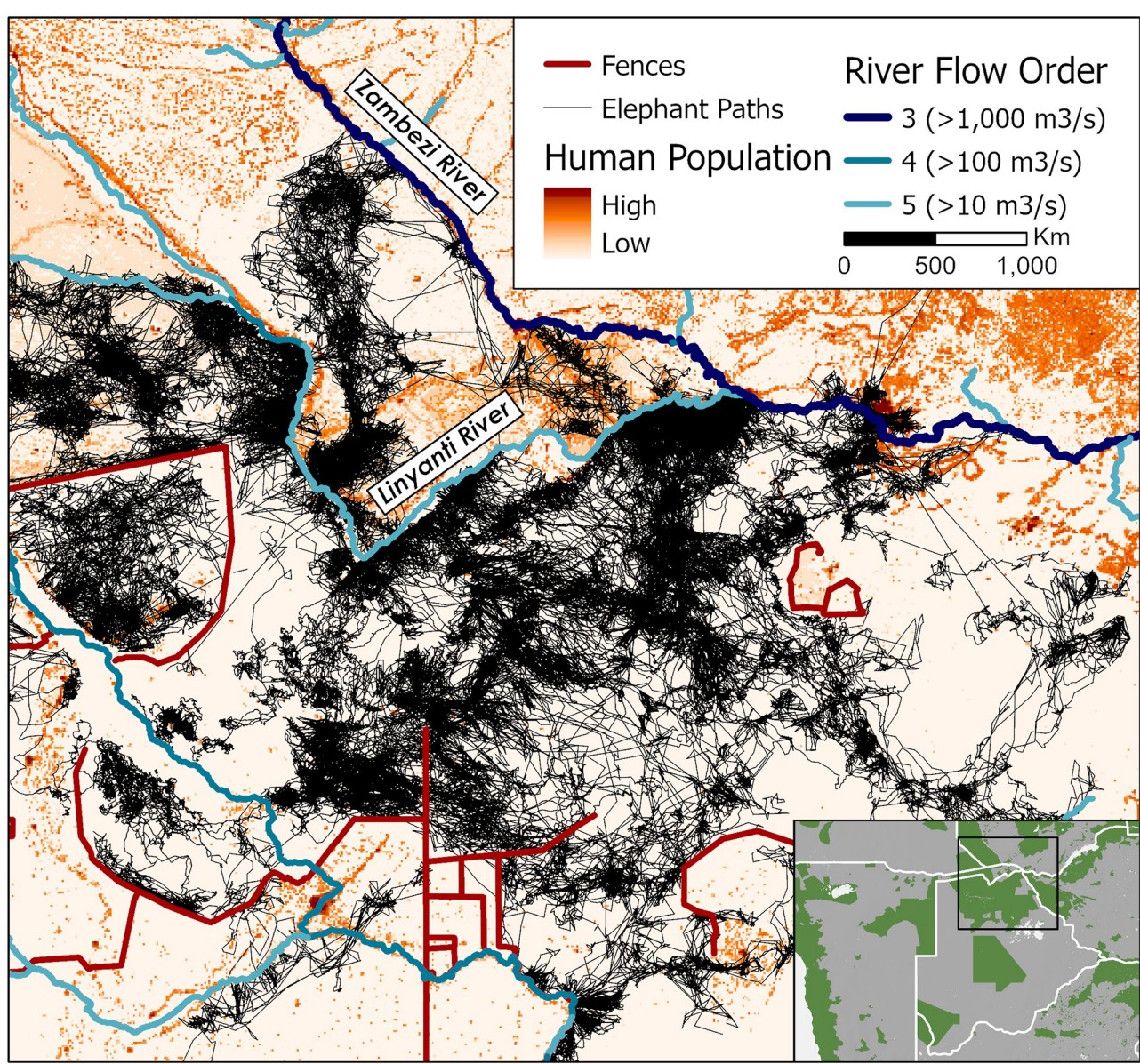

**Fig 8. Map of elephant paths and human population in the Chobe cluster.** Telemetry paths of elephants (black) in the Zambezi and northern Botswana regions overlaid on a human population density map (orange) [31]. Rivers (blue-purple lines) [30] and fences (red lines) [4] also act as barriers to elephant movement. Green areas in the subset map mark protected areas from the WDPA [35].

Okavango Delta and the Central Kalahari Game Reserve. At first glance, the transit area does not appear to be suitable habitat given the lack of surface water. Still, boreholes for cattle ranching permit long-distance movements by elephants. However, the extensive network of veterinary fences makes these routes complex. Concerted efforts to reduce fences and promote coexistence with local communities would make this region a feasible linkage.

*Greater Limpopo.* The Great Limpopo Transfrontier Conservation Area extends over 80,000 km² and includes Kruger National Park, Limpopo National Park, and Gonarezhou National Park. More than 30,000 elephants live in the area, making the region highly significant for elephant conservation. This area and that to the east towards Banhine and Zinave National parks are suitable for elephant movement due to the low human activity at present.

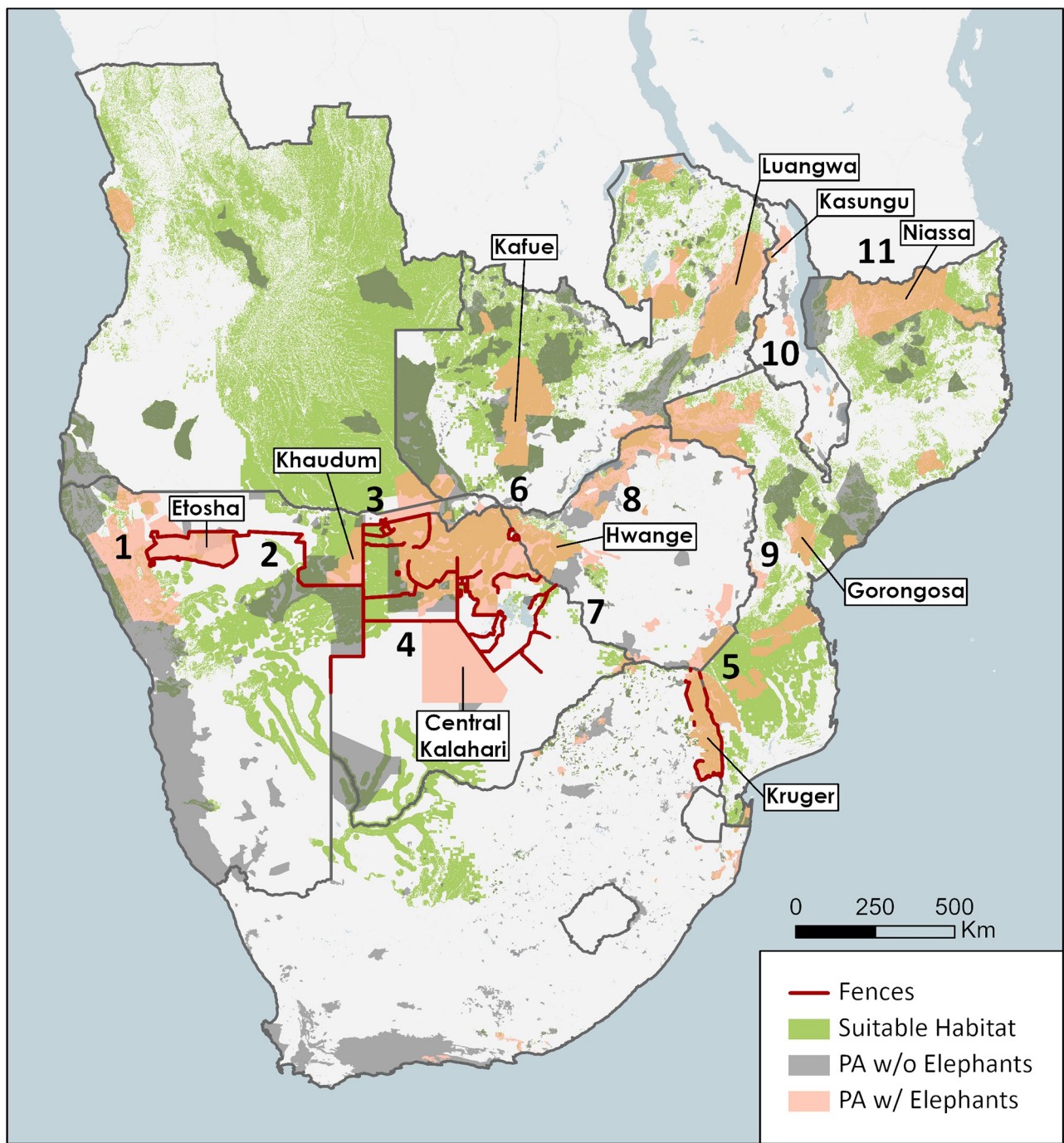

**Fig 9. Map of notable routes of connectivity or lack thereof.** Predicted suitable habitat (green), protected areas with elephants (orange), and protected areas without elephants (grey). The numbers refer to interesting connections and are described further in the main text.

Maintaining such suitable conditions and the promotion of movement amongst these parks may become an example of connectivity between unfenced protected areas.

**Zimbabwe, Zambia, Mozambique, and Malawi: Cattle, crops, and poaching create barriers.** Fences are less common throughout the remainder of southern Africa. Still, a mosaic of human settlements, cropland, and cattle rangeland significantly reduces the amount of land

suitable for elephants. What habitat remains is often in thin strips along national boundaries or completely severed into isolated blocks. Unfortunately, high incidences of poaching present a safety risk to elephants and further limit movement in several potential corridors [48].

*Northern Chobe.* While coexistence between humans and elephants is a goal of the Kavango-Zambezi Transfrontier Conservation Area [47, 49], elephant dispersal is ultimately dependent on the permeability of human settlements along the Namibia-Zambia border and major rivers, the Chobe and the Zambezi (Fig 5, [17]). Beyond the border, movement northwards towards Sioma Ngwezi National Park and Kafue National Park in Zambia is hampered by extensive rangeland. Several proposed corridors would facilitate movement across this otherwise unsuitable landscape [50, 51]. Still, they only represent the start of connectivity efforts as they do not reach the elephant population in Kafue National Park which is part of this transfrontier initiative.

*Hwange National Park.* The route to connect the Chobe cluster with the Limpopo region runs southeast from Hwange National Park along the Botswana-Zimbabwe border. Genetic evidence suggests functional connectivity between these populations [52], and elephants have been observed sporadically throughout this connection. Cattle are present throughout this area but at medium-to-low densities (5–10 cattle/km$^2$). This level of human activity is just above the threshold for which elephants may prefer (S6 Fig), and thus this region may only be marginally suitable. This linkage warrants further satellite tracking to monitor how elephants move through this human-dominated landscape along international borders.

*Northwest Zimbabwe.* Elephants occur throughout the protected areas and communal conservancies and hunting concessions that cover the northwest border of Zimbabwe and along Lake Kariba (Fig 9). This area is characterised by complicated topography and steep slopes that confine the movements of elephants in places. Unfortunately, this linkage is threatened by high levels of poaching [53]. Additionally, encroaching land conversion in these protected areas leads to a high prevalence of human-elephant conflict. As a result, the populations of elephants in this region have been decreasing [54]. Altogether, these factors limit the possible routes, yet this linkage is vital to connect the Chobe population with the populations of the Luangwa valley in Zambia.

*Central Mozambique.* North of Gonarezhou National Park, Mozambique is a highly fragmented landscape with pockets of elephants found in protected areas such as Gorongosa National Park and Gile National Reserve. However, Mozambique has seen a significant increase in human population and associated land conversion for agriculture [40], effectively isolating these elephants. Given the current trends of human activity, it is unlikely that a connection through Mozambique is feasible.

*Malawi.* Malawi is one of Africa's most densely populated countries and, as a result, restricts wildlife to protected areas. Communities and agricultural land are often immediately adjacent to these protected areas, such as Kasungu National Park. Although elephants make short forays into these anthropogenically-dominated landscapes and are the source of human-elephant conflict (Fig 6), elephants have little opportunity to make long-distance movements. Therefore, any eastwards connections are likely unfeasible, but the westwards connection towards the Luangwa may be considered. This also holds for the small population of elephants in the Nyika National Park in Malawi.

*Niassa National Reserve.* The Niassa National Reserve in northeast Mozambique is part of a transfrontier effort with the Selous in Tanzania. It contains the largest population of elephants in Mozambique. The large population and connection to the elephants of eastern Africa make this region a vital linkage to maintain, but it is threatened by high incidences of poaching [55].

## Discussion

The once contiguous savanna elephant population across southern Africa is now fragmented [56]. Populations are constrained to protected areas. Such places do not support agriculture or husbandry, and laws prevent settlement in most IUCN I-IV category protected areas. Meanwhile, favourable elephant habitat outside of protected areas has low, moderate, or heavy human influence. Our maps show the first and indicate the areas of immediate interest for elephant conservation. Then there are the areas where connectivity is broken and human activity is high enough to make new connections unfeasible. Crossing Malawi or connecting northern and southern Mozambique are such examples. The largest potential for discussion lies in the regions where moderate human impacts overlap with elephant preferences. Areas such as northern Namibia or the Botswana-Zimbabwe border are regions where elephants roam but conflict with local communities. While many of these regions represent the last opportunities to connect conservation clusters, what remains unknown is the feasibility of connections. Many communities are amenable to compensation for relocations, but many promising frameworks exist for sustainable coexistence [57]. It is this intersection of need and potential that should drive conservation efforts.

One cannot discuss connectivity throughout southern Africa without acknowledging the many fences across this landscape. Fences may benefit local communities by protecting cattle from foot-and-mouth [4, 58] or preventing human-wildlife conflict [59]. However, these benefits come at the expense of constraints on the landscape's health by artificially manipulating the abundance of local wildlife [60, 61]. These consequences include excluding critical species, elevating local population densities [27], decreased genetic variability, and overexploitation of resources. These concerns may trigger other short-term yet costly solutions such as culling or managed relocations. Specifically for elephants, fences concentrate individuals near them, depleting the vegetation [12, 62]. When barriers restrict animal movement, the consequences can be severe. Low rainfall years caused higher juvenile mortality for fenced than unfenced populations [13]. A recent event involved the deaths of 350 elephants in an 8,700 $km^2$ area of northern Botswana. Although the immediate cause of death is still uncertain, fences and other barriers prevented the animals from dispersing elsewhere [24]. Finally, when Kruger National Park was fenced entirely, managers were so concerned by high elephant numbers that they culled some 17,000 animals over 27 years [63].

Telemetry data throughout Africa show the abrupt disruption fences cause on elephant movement (Fig 3). Rather than asking the question to fence or not, the more helpful questions are how can we balance fences against connectivity? Areas like the eastern border of Kasungu National Park are apparent candidates for strong fencing, given the high potential for conflict over crop-raiding behaviour (Fig 7) and the impracticality of any natural corridor (Fig 9). Conversely, the dismantling of fences in creating transfrontier parks such as the Greater Limpopo may be effective in maintaining the ecological integrity of such landscapes. With a richer understanding of where connectivity is necessary or impractical, we may make more informed decisions about when fences are necessary and when they may be removed.

For many parts of Africa, the answer of whether it is possible to reconnect elephant populations into a more robust metapopulation is not dependent on environmental constraints nor anthropogenic ones but rather on socio-political will. Are local communities willing to move their families and livelihoods to avoid necessary elephant routes? Will governments cooperate in removing national border fences? These are difficult questions for sure, but we must pose them when seeking connectivity across a continent. The network of protected areas provides a foundation, but long-term population stability will be difficult at best without the dispersal of elephants. While some parks will experience an overabundance of

elephants that cannot disperse, others may have lost their populations to poaching or drought. In cooperation with local communities and governments, protecting the connections identified here for dispersal may represent some of our best chances at a sustainable future for elephants.

## Supporting information

**S1 Table. Telemetry sample size.** Table of the number of males and female elephants providing telemetry data broken down by conservation cluster of occurrences. Included are the major protected areas in each cluster.
(XLSX)

**S2 Table. Area of layer intersection.** Table providing amount of area considered suitable within each country of interest for each data layer used in addition to the combined data.
(XLSX)

**S1 Fig. Map of suitable landscapes.** High resolution map showing areas that are both environmentally suitable for elephants and currently experience low human activity.
(TIF)

**S2 Fig. Distribution of slope across conservation clusters.** Histogram of elephant telemetry points at various slopes for each metapopulation cluster. The red dashed line indicates the threshold (3˚) of preference for suitability.
(TIF)

**S3 Fig. Distance to rivers across conservation clusters.** Accumulation curve of area within a conservation cluster as the distance increases away from rivers of varying flow orders.
(TIF)

**S4 Fig. Distribution of crop probability across conservation clusters.** Histogram of elephant telemetry points at various cropland probabilities for each metapopulation cluster outside of protected areas. The red dashed line indicates the threshold (25%) of preference for suitability.
(TIF)

**S5 Fig. Distribution of human population density across conservation clusters.** Histogram of elephant telemetry points at various human population densities for each metapopulation cluster outside of protected areas. The red dashed line indicates the threshold (25 people per km$^2$) of preference for suitability.
(TIF)

**S6 Fig. Elephant interactions with cattle.** A) Histogram of elephant telemetry points at various cattle densities for each metapopulation cluster outside of protected areas. The red dashed line indicates the threshold (5 cattle per km$^2$) of preference for suitability. B) A map of elephant telemetry points illustrating how spill over from protected areas leads to interactions with areas of high cattle density.
(TIF)

## Acknowledgments

We acknowledge the in-kind logistical support of South African National Parks. We also thank the Gonarezhou Conservation Trust for telemetry data from Gonarezhou National Park. IFAW continues to support our ongoing research initiatives.

## Author Contributions

**Conceptualization:** Ryan M. Huang, Rudi J. van Aarde, Stuart L. Pimm.

**Data curation:** Ryan M. Huang, Rudi J. van Aarde, Michael J. Chase, Keith Leggett.

**Formal analysis:** Ryan M. Huang, Rudi J. van Aarde, Stuart L. Pimm.

**Funding acquisition:** Rudi J. van Aarde.

**Investigation:** Ryan M. Huang, Rudi J. van Aarde.

**Methodology:** Ryan M. Huang, Rudi J. van Aarde, Stuart L. Pimm, Michael J. Chase, Keith Leggett.

**Project administration:** Rudi J. van Aarde, Stuart L. Pimm.

**Resources:** Ryan M. Huang, Rudi J. van Aarde, Stuart L. Pimm.

**Software:** Ryan M. Huang.

**Supervision:** Rudi J. van Aarde.

**Validation:** Ryan M. Huang, Rudi J. van Aarde, Michael J. Chase, Keith Leggett.

**Visualization:** Ryan M. Huang.

**Writing – original draft:** Ryan M. Huang, Rudi J. van Aarde, Stuart L. Pimm.

**Writing – review & editing:** Ryan M. Huang, Rudi J. van Aarde, Stuart L. Pimm, Michael J. Chase, Keith Leggett.

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
