## [Decision Letter · Decision Letter 0]

21 Jul 2022

PONE-D-22-17386Defragmenting Southern Africa’s elephant populationsPLOS ONE

Dear Dr. Huang,

Thank you for submitting your manuscript to PLOS ONE. After careful consideration, we feel that it has merit but does not fully meet PLOS ONE’s publication criteria as it currently stands. Therefore, we invite you to submit a revised version of the manuscript that addresses the points raised during the review process.

We look forward to receiving your revised manuscript.

Kind regards,

Bi-Song Yue, Ph.D

Academic Editor

PLOS ONE

Journal Requirements:

"Support for this study was provided by Billiton, Conservation Foundation Zambia, Conservation International’s southern Africa’s Wildlife Programme, the Conservation Lower Zambezi, the International Fund for Animal Welfare, the Mozal Community Development Trust, the National Research Foundation, the National Postcode Lottery of the Netherlands, Peace Parks Foundation, the US Fish and Wildlife Services, the University of Pretoria, the World Wildlife Fund (SARPO; Mozambique; SA), the Walt Disney Grant Foundation, and the Wildlifewins Lottery. Elephants Without Borders was funded by the Paul G. Allen Family Foundation, Jody Allen, the Woodtiger Fund, the Thomas C Bishop Charitable Foundation, the James and Deborah Burrows Foundation, and the Zoological Society of San Diego. "

"Support for this study was provided by Billiton, Conservation Foundation Zambia, Conservation International’s southern Africa’s Wildlife Programme, the Conservation Lower Zambezi, the International Fund for Animal Welfare, the Mozal Community Development Trust, the National Research Foundation, the National Postcode Lottery of the Netherlands, Peace Parks Foundation, the US Fish and Wildlife Services, the University of Pretoria, the World Wildlife Fund (SARPO; Mozambique; SA), the Walt Disney Grant Foundation, and the Wildlifewins Lottery. Elephants Without Borders was funded by the Paul G. Allen Family Foundation, Jody Allen, the Woodtiger Fund, the Thomas C Bishop Charitable Foundation, the James and Deborah Burrows Foundation, and the Zoological Society of San Diego. We acknowledge the in kind logistical support of South African National Parks. This research was sanctioned and supported by the Botswana Dept. of Wildlife & National Parks, Direcção Nacional de Areas de Conservação, the Namibian Ministry of Tourism & Environment, the Malawian Wildlife Dept., the South African National Parks, Ezemvelo KZN Wildlife of South Africa, and the Zambian Wildlife Authority. We also thank the Gonarezhou Conservation Trust for telemetry data from Gonarezhou National Park.  Ifaw continues to support our ongoing research initiatives.

The funders had no role in study design, data collection and analysis, decision to publish, or preparation of the manuscript"

5. We note that Figures 2, 4,5,6,8,9 and 10 in your submission contain [map/satellite] images which may be copyrighted. All PLOS content is published under the Creative Commons Attribution License (CC BY 4.0), which means that the manuscript, images, and Supporting Information files will be freely available online, and any third party is permitted to access, download, copy, distribute, and use these materials in any way, even commercially, with proper attribution. For these reasons, we cannot publish previously copyrighted maps or satellite images created using proprietary data, such as Google software (Google Maps, Street View, and Earth). For more information, see our copyright guidelines: http://journals.plos.org/plosone/s/licenses-and-copyright.

a. You may seek permission from the original copyright holder of Figures 2, 4,5,6,8,9 and 10 to publish the content specifically under the CC BY 4.0 license.  

Reviewers' comments:

Reviewer's Responses to Questions

**Comments to the Author**

1. Is the manuscript technically sound, and do the data support the conclusions?

Reviewer #1: No

Reviewer #2: Partly

2. Has the statistical analysis been performed appropriately and rigorously? 

Reviewer #1: No

Reviewer #2: I Don't Know

3. Have the authors made all data underlying the findings in their manuscript fully available?

Reviewer #1: No

Reviewer #2: No

4. Is the manuscript presented in an intelligible fashion and written in standard English?

Reviewer #1: No

Reviewer #2: Yes

5. Review Comments to the Author

Reviewer #1: PONE-D-22-17386: Defragmenting Southern Africa’s elephant populations

One of the main strengths of this work lies in the amount of telemetry and environmental/anthropogenic data collected over space and time by the authors, although as they are presented in the methodology, they do not explicitly indicate which research questions/objective they intend to answer with these data. For example, lines 97-106 are describing data acquisition from various sources but which objectives (which should have been clearly stated in the introduction section) are they addressing?

The statement in the “lines 110, 143, 164&254” for example, should have been used to guide the formulation and write up of this manuscript-understanding suitable habitats for elephants in southern Africa and how artificial barriers from different areas are limiting their connectivity and how to respond to these barriers/challenges, use lessons from these to set best practices that are context based?-unfortunately the paper as it stands, it does not reflect that this is the case.

One major flaw of the paper is the inconsistencies in writing and presentation of information-in the course of reading this, it was difficult to follow clearly from the title, introduction, methods, what exactly authors were trying to address….for example, one would ask, what is the objective of the paper? Is it about what the titles states? If so, one would have expected authors to set the scene and the aim of the paper, right from the introduction, however, this seems rather missing.

Reservations in the specific sections of the paper

Title

Despite the being rather ambitious, it would rather have some more action words on how realistic this can be given the current rate of local and global changes, that include among others climate and human population change and associated pressures

Abstract

The abstract seems a little twisted into a more descriptive than a conventional scientific way of presenting a research work with clear aim, methods results and conclusion…

Introduction

It would be good to see this section explaining explicitly the clear goals of the paper so readers can follow how this resonates with the subsequent sections. Also, authors could mention and describe sooner in the manuscript a sentence about the elephants populations being presented in this paper and cite the relevant figure (e.g. Fig.2) as well as a mention of the major threats that each of this is facing and the way defragmenting process would help address/mitigate these threats

Lines 80-84: It would be good to state if these hypothesis and based on which assumptions? And how are they linked with the objectives of the paper?

Methods’ section

Being a research article, it would be better to state sampling (data collection) strategy, sample size (e.g. source and number of individual elephants collared in each population, number of protected areas involved in data collection), data collection and analysis (including those other than collaring data) separately, rather than the present arrangement in the methodology section. For example, one would want to know, which sections in the “methods-lines 85-364” are methods and which ones are results? And what do all case studies in this section represent?-they are not stated anywhere in the introduction section and in the methodology it is not clear why they are being presented.

Which research question/objective do long-term telemetry data (lines 86-96) intend to answer in this paper? Is this backed up by any background hypothesis in the introduction section of the paper? These should have come a bit earlier in the introduction section of the paper so they guide the subsequent sections. For example, if questions in lines 110, 143&254 came out clearly in the introduction section, then the readers would be able to follow easily the flow of the story of this paper. Without this, it is difficult to have a meaningful interpretation and discussion of the results

What is SRTM in line 99?

There is no clear link between the subsequent sections of the paper (after introduction section) and the title of this paper. For example, not sure how one would expect elephants to cross fenced areas and how this would be resolved by telemetry data to achieve the ultimate goal which is reconnecting the populations.

I have looked at the manuscript and found a number of issues and problems related to the scientific clarity. Clearly the paper doesn't seem to state clear objective and rationale, and the methodology is not well formulated and implement in accordance to the title of the paper which makes interpretation of the synthesized information difficult to readers.

In view of the comments above, I am afraid that I cannot recommend this manuscript for publication in PLOS ONE.

Reviewer #2: The authors present and interesting and important topic of relevance to the conservation and management of elephants across Africa. The need for this type of analysis is clear and important due to the large spatial requirements of elephants. However, I do find that the method section should be extended so that the methods used could be replicated. Very little information is provided on some technical details such as the degrees of overlap between the layers. Some figures are also confusing. For example, Figure 3 shows for those few humans (D) cover the entire southern regions (omitting Eswatini and Lesotho and this should also be stated and the reasons as Eswatini does have elephants). If this is the underlying layer as presented, it would seem so course and poorly defined that I wonder how it can be meaningful. Also, as an example the ‘few’ category for cattle is described in a contradictory manner in the text and is also poorly defined:

Line 329-330, Page 14

Cattle are present throughout this area but at relatively low densities (5-10/km2).

Line 152, Page 7

The same pattern holds if one considers areas with high human populations (>25 people/km2) [37] and high cattle densities (>5 cattle/km2) (S2B Fig) [38].

Other details are also inaccurate or not updated. As example Banhine- and Zinave National Park in Mozambique are shown as without elephants in figure 10 but both do have elephants. The latter due to a recent translocation. Again, this classification may be due to definitions that are not well explained in the text of figure headings.

Overall, although the topic and analyses are important, the lack of clarity in terms of definitions of categories and methods, other than broadly stating how the overlap in layers were used, need more work before the manuscript can be published.

Minor Grammatical corrections:

Line 102 Page 5

‘extend’ not ‘extent’

Line 314-315, page 14

Unfortunately, high incidences of poaching place further pressures that endanger movement in several

potential connections [53].

Rewrite

Unfortunately, high incidences of poaching present a safety risk to elephants and further limit movement in several potential corridors

Line 371, page 16

Preferably change ‘infeasible’ to ‘unfeasible’

6. PLOS authors have the option to publish the peer review history of their article (what does this mean?). If published, this will include your full peer review and any attached files.

Reviewer #1: No

Reviewer #2: No

---

## [Author Response · Author response to Decision Letter 0]

12 Sep 2022

PONE-D-22-17386: Defragmenting Southern Africa’s elephant populations

One of the main strengths of this work lies in the amount of telemetry and environmental/anthropogenic data collected over space and time by the authors, although as they are presented in the methodology, they do not explicitly indicate which research questions/objective they intend to answer with these data. For example, lines 97-106 are describing data acquisition from various sources but which objectives (which should have been clearly stated in the introduction section) are they addressing? 

The objectives were listed in the last paragraph of the Introduction (lines 74-84). We have now revised the Introduction for clarity. We now also more explicitly link our data acquisition to the objectives at the end of the Introduction.

The statement in the “lines 110, 143, 164&254” for example, should have been used to guide the formulation and write up of this manuscript-understanding suitable habitats for elephants in southern Africa and how artificial barriers from different areas are limiting their connectivity and how to respond to these barriers/challenges, use lessons from these to set best practices that are context based?-unfortunately the paper as it stands, it does not reflect that this is the case.

The questions on lines 110, 143, 164, & 254 are derived from the original Introduction, specifically lines 80-84. We have now revised the Introduction for clarity.

One major flaw of the paper is the inconsistencies in writing and presentation of information-in the course of reading this, it was difficult to follow clearly from the title, introduction, methods, what exactly authors were trying to address….for example, one would ask, what is the objective of the paper? Is it about what the titles states? If so, one would have expected authors to set the scene and the aim of the paper, right from the introduction, however, this seems rather missing.

Our objective was stated in Line 81: “Our objective is to map where elephants might be able to move between their current populations and, conversely, where it is unlikely that they can do so.” To address the reviewer’s confusion, we have now revised the title and wording in the Introduction. The Introduction sets the scene by first explaining the need of restoring connections in a fragmented landscape, then describing the fragmented nature of elephant populations and its causes, and ending with our objectives and approach to solve this problem. We also expanded the existing description of our approach at the end of the Introduction.

Reservations in the specific sections of the paper

Title

Despite the being rather ambitious, it would rather have some more action words on how realistic this can be given the current rate of local and global changes, that include among others climate and human population change and associated pressures 

We have now changed the title to match the paper’s objectives more explicitly

Abstract

The abstract seems a little twisted into a more descriptive than a conventional scientific way of presenting a research work with clear aim, methods results and conclusion…

We have reviewed the Abstract and ensured that it follows the aim (now L24-25), the methods (now L25-28), our results (now L28-30) and our conclusions (now L31-33).

Introduction

It would be good to see this section explaining explicitly the clear goals of the paper so readers can follow how this resonates with the subsequent sections. Also, authors could mention and describe sooner in the manuscript a sentence about the elephants populations being presented in this paper and cite the relevant figure (e.g. Fig.2) as well as a mention of the major threats that each of this is facing and the way defragmenting process would help address/mitigate these threats

We have now rewritten the Introduction to provide more explicit background and explicitly state the goals of the paper.

Lines 80-84: It would be good to state if these hypothesis and based on which assumptions? And how are they linked with the objectives of the paper?

We have greatly expanded our explanation in the newly revised Introduction (now L55-82).

Methods’ section

Being a research article, it would be better to state sampling (data collection) strategy, sample size (e.g. source and number of individual elephants collared in each population, number of protected areas involved in data collection), data collection and analysis (including those other than collaring data) separately, rather than the present arrangement in the methodology section. For example, one would want to know, which sections in the “methods-lines 85-364” are methods and which ones are results? 

We initially tried to separate the Methods Section from the “Where do elephants want to go?” and “How human actions restrict elephant movements” into separate sections using the PLOS One level one heading, but realize this was not clear. We have now made the Results section more explicit and lowered the two sections to level two headers. Lastly, we now include a Supplemental table that includes sample sizes for each cluster and the relevant protected areas.

And what do all case studies in this section represent?-they are not stated anywhere in the introduction section and in the methodology it is not clear why they are being presented.

We have revised the Introduction to clarify the inclusion of the case studies.

Which research question/objective do long-term telemetry data (lines 86-96) intend to answer in this paper? Is this backed up by any background hypothesis in the introduction section of the paper? These should have come a bit earlier in the introduction section of the paper so they guide the subsequent sections. 

We have now explicitly mentioned how the use of telemetry data is linked to our objectives in the Introduction.

For example, if questions in lines 110, 143&254 came out clearly in the introduction section, then the readers would be able to follow easily the flow of the story of this paper. Without this, it is difficult to have a meaningful interpretation and discussion of the results

As answered above, the questions on lines 110, 143, & 254 were straight from the Introduction, specifically lines 80-84 (now L59, 64, 72, and 75). 

What is SRTM in line 99?

SRTM stands for Shuttle Radar Topography Mission, a common dataset used by many GIS users. We have now removed the acronym given that the citation is present.

 There is no clear link between the subsequent sections of the paper (after introduction section) and the title of this paper. For example, not sure how one would expect elephants to cross fenced areas and how this would be resolved by telemetry data to achieve the ultimate goal which is reconnecting the populations.

The revised manuscript now better explains the role of telemetry data in determining elephant habitat preferences and demonstrating the impact of fences. We also now mention our use of telemetry data in the Discussion when discussing the role of fences.

I have looked at the manuscript and found a number of issues and problems related to the scientific clarity. Clearly the paper doesn't seem to state clear objective and rationale, and the methodology is not well formulated and implement in accordance to the title of the paper which makes interpretation of the synthesized information difficult to readers.

In view of the comments above, I am afraid that I cannot recommend this manuscript for publication in PLOS ONE.

We have now revised the title, Introduction, and Methods to be clearer in our approach.

Reviewer #2: The authors present and interesting and important topic of relevance to the conservation and management of elephants across Africa. The need for this type of analysis is clear and important due to the large spatial requirements of elephants. However, I do find that the method section should be extended so that the methods used could be replicated. Very little information is provided on some technical details such as the degrees of overlap between the layers. 

We have expanded the methods section to be more explicit in our process and also included a supplemental table that lists the exact amount of overlap area for each data layer and country.

Some figures are also confusing. For example, Figure 3 shows for those few humans (D) cover the entire southern regions 

The issue with Fig 3D was due to how raster data are displayed when zoomed out. We have corrected the figure so it now displays properly (now Fig 1).

(omitting Eswatini and Lesotho and this should also be stated and the reasons as Eswatini does have elephants).

We initially excluded Eswatini and Lesotho given the low numbers of elephants (none occur in Lesotho). In response to the reviewer, we have now included Eswatini in the analysis, but have included a statement on the exclusion of Lesotho.

If this is the underlying layer as presented, it would seem so course and poorly defined that I wonder how it can be meaningful. 

The resolution of the human population data occurs at a 1km scale, which we believe provides a sufficient resolution for continental and regional mapping. We believe the problem here is one of visualization rather than the underlying data. We have addressed this by revising Fig 3D (now Fig 1D).

Also, as an example the ‘few’ category for cattle is described in a contradictory manner in the text and is also poorly defined:

Line 329-330, Page 14

Cattle are present throughout this area but at relatively low densities (5-10/km2).

Line 152, Page 7

The same pattern holds if one considers areas with high human populations (>25 people/km2) [37] and high cattle densities (>5 cattle/km2) (S2B Fig) [38].

We thank the reviewer for pointing out this discrepancy. We have now revised the text to be more consistent.

Other details are also inaccurate or not updated. As example Banhine- and Zinave National Park in Mozambique are shown as without elephants in figure 10 but both do have elephants. The latter due to a recent translocation. Again, this classification may be due to definitions that are not well explained in the text of figure headings.

Again, we thank the reviewer for this comment. It appears that our data on protected elephants was out of date and have updated the map to correctly reflect the inclusion of elephants in Banhine and Zinave.

Overall, although the topic and analyses are important, the lack of clarity in terms of definitions of categories and methods, other than broadly stating how the overlap in layers were used, need more work before the manuscript can be published.

We have now addressed all of the definitions and methods the Reviewer has mentioned. We also include a new table in the supplemental material that provides exact numbers of overlap for each underlying layer.

Minor Grammatical corrections:

Line 102 Page 5

‘extend’ not ‘extent’

‘extent’ was the intended here but we dropped “range from”. Thank you for pointing this out.

Line 314-315, page 14

Unfortunately, high incidences of poaching place further pressures that endanger movement in several

potential connections [53].

Rewrite

Unfortunately, high incidences of poaching present a safety risk to elephants and further limit movement in several potential corridors

We have made the suggested change.

Line 371, page 16

Preferably change ‘infeasible’ to ‘unfeasible’

We have also made this change.

---

## [Editor Report · Decision Letter 1]

26 Sep 2022

Mapping potential connections between Southern Africa’s elephant populations

PONE-D-22-17386R1

Dear Dr. Huang,

We’re pleased to inform you that your manuscript has been judged scientifically suitable for publication and will be formally accepted for publication once it meets all outstanding technical requirements.

Kind regards,

Bi-Song Yue, Ph.D

Academic Editor

PLOS ONE

---

## [Editor Report · Acceptance letter]

29 Sep 2022

PONE-D-22-17386R1 

Mapping potential connections between Southern Africa’s elephant populations 

Dear Dr. Huang:

I'm pleased to inform you that your manuscript has been deemed suitable for publication in PLOS ONE. Congratulations! Your manuscript is now with our production department. 

Kind regards, 

on behalf of

Dr. Bi-Song Yue 

Academic Editor

PLOS ONE